# Analysis of the Dynamic Height Distribution at the Estuary of the Odra River Based on Gravimetric Measurements Acquired with the Use of a Light Survey Boat—A Case Study

**DOI:** 10.3390/s20216044

**Published:** 2020-10-23

**Authors:** Krzysztof Pyrchla, Arkadiusz Tomczak, Grzegorz Zaniewicz, Jerzy Pyrchla, Paulina Kowalska

**Affiliations:** 1Faculty of Electronics, Telecommunications and Informatics, Gdańsk University of Technology, 80-233 Gdańsk, Poland; krzpyrch@student.pg.edu.pl; 2Faculty of Navigation, Maritime University of Szczecin, 70-500 Szczecin, Poland; g.zaniewicz@am.szczecin.pl; 3Faculty of Civil and Environmental Engineering, Gdańsk University of Technology, 80-233 Gdańsk, Poland; paulina.kowalska@pg.edu.pl

**Keywords:** gravity anomalies and earth structure, gravimetric river survey, geophysical river survey, Fourier analysis, numerical modelling, time-series analysis, Europe

## Abstract

This article presents possible applications of a dynamic gravity meter (MGS-6, Micro-g LaCoste) for determining the dynamic height along the Odra River, in northwest Poland. The gravity measurement campaign described in this article was conducted on a small, hybrid-powered survey vessel (overall length: 9.5 m). We discuss a method for processing the results of gravimetric measurements performed on a mobile platform affected by strong external disturbances. Because measurement noise in most cases consists of signals caused by non-ideal observation conditions, careful attempts were made to analyze and eliminate the noise. Two different data processing strategies were implemented, one for a 20 Hz gravity data stream and another for a 1 Hz data stream. A comparison of the achieved results is presented. A height reference level, consistent for the entire estuary, is critical for the construction of a safe waterway system, including 3D navigation with the dynamic estimation of under-keel clearance on the Odra and other Polish rivers. The campaign was conducted in an area where the accuracy of measurements (levelling and gravimetric) is of key importance for shipping safety. The shores in the presented area of interest are swampy, so watercraft-based measurements are preferred. The method described in the article can be successfully applied to measurements in all near-zero-depth areas.

## 1. Introduction

Poland’s region of river estuaries is flat; therefore, the river gradients are also small. The vertical reference network in the region of the main Polish rivers, the Odra and the Wisła (Vistula), is necessary for hydrographic, hydrological and hydrodynamic surveys and would have a great impact on developing 3D vessel navigation systems with the dynamic determination of under-keel clearance. The network can also serve for developing a regional digital terrain model. Today, gravity missions in space are an important tool for obtaining global data by providing coverage all over the world [1]. In large estuaries, created vertical reference networks are based on satellite data. With regard to the estuaries of rivers flowing into the Baltic Sea, the resolution of satellite data is insufficient [2] to provide the basis for an analysis of vertical reference networks. Additional data are usually required in such areas to ensure regional geopotential models of acceptable spatial resolution [3]. These data can be obtained from local accurate measurements of gravity [4,5].

An interesting challenge is the estimation of the height in the inland water bodies, such as rivers and lakes. If the whole area of interest is accessible from land, the solution may be a land gravimetric campaign [6]. However, the diversity of environmental conditions in the area of natural inland water bodies can lead to a situation where significant parts of the area of interest are impossible to reach for land measurements. A practical solution is to employ a research vessel, but such a vessel must have a small draft and high maneuverability for operation in a swampy river estuary, which are conditions that reduce the maximum size of such a vessel. During our measurements, gravimetric equipment was installed on a small survey vessel, 9 m in length.

A small vessel engaged in gravimetric measurements has to be prepared for the measurement itself and for retrieving the signal from noise generated by the environment [5,7]. One possible solution is the application of a next-generation strapdown marine gravimeter, which is able to achieve accuracy better than 1 mGal and can perform measurements on curved profiles [8,9]. Another solution could be the GNSS (Global Navigation Satellite Systems) estimation of the sea surface height and the use of the proper corrections to calculate the physical height [3]. In this work, we focused on another solution, namely, the application of a standard marine gravimetry system for measurements within a river estuary. This is not a common operation, because marine gravity sensors are generally sensitive to rapid platform motions [10]. This requires the development of measurement technology in which the maximum possible number of sources interfering with the measurements are eliminated. This objective is adopted in all gravimetric dynamic measurements, where a ship or aircraft provides the platform [11,12]. We can assume that the level of precision in such cases is determined by the manufacturer of the measuring device, and this level should be adopted as a reference value. The manufacturer Micro-g LaCoste indicates a measurement accuracy of 1 mGal [13], but this refers to ships more than 40 m in length. However, after all the necessary dynamic corrections are made, the accuracy of the measured gravimetric signal can be improved even beyond the general limits indicated by the manufacturer [13].

Filtration is one of the key elements of dynamic gravimetric data processing. In this analysis, it was decided to use a low-pass fast Fourier transform (FFT) filter, as described in [14]. Parameters such as the cut-off frequency and transition band were chosen following the spectral analysis of the collected data. The spectral analysis included horizontal accelerations of the vessel to determine in which parts of the band their energy was concentrated. During data analysis, the ship’s accelerations were found to be the main source of the noise in the recorded data. Therefore, we focused on selecting frequency filters that would reduce the impact of accelerations as much as possible.

The standard procedure for verifying the consistency of gravimetric measurement data is to determine their internal accuracy [15,16]. Such accuracy is verified by performing part or all measurements on a survey line at least twice and statistically analyzing the obtained differences.

However, such procedures do not provide information on external data accuracy. To determine the value of the absolute error, it is not sufficient to analyze internal data consistency only, but a comparison of the results with reliable data from an independent campaign must also be made. Based on this assumption, a land-based measurement campaign was conducted along the riverbank. A CG-5 gravimeter from Micro-g LaCoste was used for the measurements. A representative group of data gathered from the campaign enabled reliable comparisons. The adopted measurement methodology allows for the determination of the internal and external accuracies of the measurement data so that the filtration effectiveness can be assessed. 

This article presents experimental gravimetric research carried out on a river, in northwest Poland. The work consisted of taking high-resolution gravimetric images of increased density and achieving better accuracy for the measurements. We determined the character of the gravitational acceleration field in inland waters of the Odra River mouth, where previously, such an accurate and professional gravimeter had not been used. The results of the measurements were used for modelling variations in the dynamic surface of the river. The system was also used to improve the technology of gravimetric data analysis referring to areas of so-called zero depth.

The rest of the paper proceeds as follows: Section 2 provides a description of the experimental setup and data processing methods, Section 3 presents the results achieved using the described methods, and the Section 4 contains some brief conclusions based on our findings.

## 2. Materials and Methods 

The dynamic marine gravity meter MGS-6 (Micro-g LaCoste, Lafayette, CO 80026, USA) from LaCoste & Romberg—Scintrex Inc., a sixth-generation Marine Gravity System, was used for recording changes in gravity during the campaign. Its basic advantage is the capability of performing gravimetric measurements on a mobile platform. The MGS-6 guarantees very accurate measurements of gravitational acceleration (gravity force). The system, based on a TAGS-6 platform, has a frame supporting a gimbal and a sensor. Vibrations are damped by a gimbal, suspension strings and air-filled shock absorbers. The gimbal holds a gravity meter sensor and keeps it horizontal when the system is moving. The gravity meter sensor contains a gravity-detecting element, an oven and an electronic unit of the platform. The system is operated from a laptop computer, which also records gravity data.

The position of the research vessel was measured using two GNSS antennas and receivers, a Trimble R8 and a Leica GS 15. The SMC IMU-10 was used as an inertial sensor. The data flow in the whole measurement setup is shown in Figure 1. The physical location of each sensor is indicated in Figure 2a.

The manufacturer of the MDS-6 marine gravity measurement system recommends installation on vessels more than 40 m in length. We therefore decided to determine what results would be possible to achieve after the installation of this system on a small ship of 9 m in length. The vessel chosen for the measurements was a Hydrograf XXI. Despite its small size, this research boat has one advantage—it has hybrid propulsion (gasoline–electrical). As a result of battery pack installation, the center of gravity (COG) is lowered and the overall mass is increased in comparison to boats without this modification. During measurements, gasoline propulsion was used, but the additional mass allowed rapid hull oscillation to be avoided. Furthermore, it was assumed that the mobilization of the complete navigation sensor setup and filtration of registered signals allowed reliable and interesting results to be achieved.

For the gravimetric sensor installation, thick chipboard plates were used. The installation project used two layers of plates. The first one was fitted to the deck of the ship, so it would not slide along it. The second layer was the base to which the frame of the MGS-6 was screwed. The thickness of the plates, which was 38 mm, allowed us to mill holes for mounting screw heads. As a result, both plates adhered to each other over a large area. The other benefit of this setup is that the sensor frame could be placed in the best location before the top plate was screwed with wood screws to the bottom one. This configuration is rigid and allowed us to precisely set the points of offset measurements using geodetic techniques. The whole structure provided a solid mount for the gravimeter frame on the ship despite the lack of pre-designed mounting places on the deck. Figure 3 shows the relative location of the gravimeter frame on the mounting plates.

Dynamic gravimetric measurements are closely related to accurate inertial measurements. Sea measurements are known to have high accuracy and resolution for the acquired data [17] compared with other types of dynamic gravimetric measurements (satellite-, aviation- [18] or altimetry-based methods). The quality of the data obtained is inherently dependent on the stability of the measurement platform, because such measurements involve recording kinematic parameters of the platform transporting the gravity meter. Theoretically, all accelerations of the platform will be recorded by the measurement unit of the gravity meter. In practice, each dynamic gravity meter features a finite passband width, so it can record accelerations in a certain range of frequencies.

The MGS-6 gravimetric system can record gravity data with frequencies of 1 Hz or 20 Hz. For regular marine gravity data collection on large-scale areas of water, 1 Hz sampling is sufficient, so by default, the device software is configured to measure with this frequency. We decided to perform two campaigns on the same area using two different configurations. In the first campaign, the gravimeter was set to measure at a 1 Hz sampling rate, and during the second, it was set to 20 Hz.

Despite the slow speeds used by the survey ships and low variations in heading and speed during measurements (compared to those for airborne platforms), the accelerations onboard a vessel navigating along a survey line have an amplitude a few orders of magnitude greater than the desired anomalies in the gravitational field. A typical solution to that problem is an assumption that these disturbances occupy a band of signals with periods significantly different from those of the expected signals coming from variations in the gravitational field. For instance, the period of vibrations caused by sea waves ranges from 4 to 15 s [19]. This allows us to assume that after using a frequency filter of any type used in marine gravimetry [14,20,21], the impact of the vibrations will be eliminated from the signal.

Taking for granted such an approach to the filtration of gravimetric data is currently considered to be unjustified. While the accuracy of sea gravimetric systems continues to increase, there is a demand for increasingly higher-quality data with a low margin of uncertainty [22]. In modern dynamic gravimetry, it is crucial to accurately determine the kinematic parameters of the mobile platform. Unlike in airborne gravimetry, these parameters do not have to be processed immediately to obtain corrections for acceleration. To determine the corrections precisely, we need a vessel with an accurate positioning system and knowledge of the gravity meter pulse response. An indirect approach can be taken in which a spectral analysis of the collected kinematic data is performed to determine the band comprising the disturbances from accelerations, and then identify the optimal method of filtration [15].

The authors attempted to perform dynamic gravimetric measurements on the Odra River near its mouth. The depths necessitated a smaller survey vessel, and the measurements had to be conducted on substantially shorter lines. The area of measurements included Szczecin’s urban surroundings, introducing additional difficulty due to increased vessel traffic that was not stopped at the time of measurements.

For these reasons, the data analysis started with a detailed review of the possible disturbances of the measurement results. First, we analyzed the influence of a large-scale curvature of the measurement trajectory. The trajectory curvature was calculated from survey vessel positions. Outliers were removed from position data obtained from a differential global positioning system (DGPS), and the gaps were patched using an autoregression model. Then, the data were decimated to a spatial resolution of 35 m. The distance between the measurement points to which the data were decimated was selected by examining the numerical convergence of the curvature determination method. Based on Equation (1), the data were then used to calculate the trajectory curvature “*k*”.
(1)$$k=\frac{\left|y^{\prime \prime }x^{\prime }-y^{\prime }x^{\prime \prime }\right|}{{\left(x^{\prime^{2}}+y^{\prime^{2}}\right)}^{3/2}}$$

It is assumed in the formula that the curve is described by a parametric system of equations of the co-ordinates *x*(*t*) and *y*(*t*). In Equation (1), *x*′ and *y*″ denote the first derivatives of the parameter, and *x*″ and *y*″ denote the second derivatives.

With the value of the curvature thus calculated, its impact on the gravimetric measurement results was examined. The value of the centrifugal acceleration on the trajectory was determined directly from the trajectory curvature value. It was assumed that the long-term impact of such acceleration would be similar to that of an unlevelled (tilted) gravity meter. Based on the formula for small roll corrections [13], the value of the correction for the trajectory curvature was calculated. The results are shown in Figure 4 and Figure 5.

This correction exhibited very small values and did not exceed 1 mGal. This was caused by the small curvature of the vessel’s trajectory on the survey line and slow linear speed of approximately 2.5 m/s. Despite that, this correction was added at the final stage of data analysis.

The Eotvos correction is extremely important for dynamic measurements. The Eotvos effect may yield very high values—tens of mGal in marine gravimetry and hundreds of mGal in airborne gravimetry. The value of the Eotvos correction *g_e_* can be calculated from Equation (2):(2)$$g_{e=\;}2\;\omega \;V\;\cos\left(Lat\right)\sin\left(A\right)+\frac{1}{r}\;V^{2}$$

In Equation (2), *V* denotes the velocity of the gravity meter movement relative to the Earth, *A* is the direction of the velocity vector, *Lat* is the geographical latitude at the moment of measurement, *ω* is the rotation angular velocity of the Earth and *r* denotes the Earth’s radius.

Clearly, vessel heading and speed stability are vital. When these requirements are maintained, the correction will not vary much and can be calculated from Equation (2). To ensure high accuracy for the speed and heading measurements, the kinematic parameters of the survey vessel were measured using three devices, namely, two GPS receivers and one inertial measurement unit. The acquired data were cleaned by removing outlier data and merged in postprocessing.

Another important source of disturbance, namely, the horizontal and vertical acceleration of the survey vessel, results from the local curvature of the trajectory and the vessel’s yawing and speed variations. An attempt was made to determine in which regions of the band the concentrated energy of these oscillations was present. This was done by examining data from the dilatometers installed in the measurement sensor of the MGS-6. Example values of the acceleration recorded by these systems are shown in Figure 6.

The spectral power density was calculated for signals from both dilatometers. The results are shown in Figure 6.

An analysis of the data gathered by the dilatometers shows that the energy of the vibrations on the longitudinal axis was concentrated around the 10 s band. Therefore, it was concentrated in a band distant from the one where gravimetric signals are found. These vibrations are also less important because the values of the accelerations on this axis and the calibration constant provided by the manufacturer allowed us to calculate the vertical cross-coupling (VCC) correction, which was added before filtration and eliminated the cross-coupling effect on that axis.

In the case of vibrations on the cross axis, accelerations were found in a much broader band and reached 500 s. In arm-spring gravimeters, these accelerations, theoretically, are not coupled with the measurement result [19] but can cause the gimbal unit to swing, thus disturbing measurements if they have a sufficiently high amplitude and low frequency.

Therefore, having analyzed the spectral data, we decided to use a low-pass FFT filter with a cut-off period of 500 s. The width of the passband was 200 s. The low-pass FFT filter implementation, as described in the literature [14], was used for frequency filtration. This type of filter enables better separation of the signal, mainly because it is easier, compared to the case of a Butterworth filter, to control the cut-off frequency and the passband width. Consequently, the signal band with noise detected in a spectral analysis can be cut off more precisely.

In addition, the gravimetric signals recorded on the survey lines were smoothed using a second-order Savitzky–Golay filter with a window length of 500 s. It was required in order to eliminate residual oscillations in the data after frequency filtration and because there was no need for a higher spatial resolution, which was already limited by the selected frequency filtration parameters.

Using the conclusions from the first campaign, the second campaign was designed. The gravimeter was configured to record the gravity at a higher sampling rate equal to 20 Hz. This is the highest sampling frequency available in this model of gravimeter. The kinematic data have to be sampled faster with the increase in gravity data sampling, so the 10 Hz data acquisition frequency was used.

The higher sampling frequency allowed us to utilize a different type of data processing (Figure 7). The raw gravity data were corrected by the application of both the VCC correction and the Eotvos correction calculated from the navigation data. In the second stage, the resulting corrected gravity was carefully examined in order to remove all outliers from the data, after which the data were filled using the autoregressive model (ARMA). The resulting data were filtered using the same one-pass low-pass FFT that was used for processing the data from the first campaign. The cut-off wavelength was set to 600 s, and the stopband, to 400 s. No additional filtration was used.

## 3. Results

In this work, the data collected in two campaigns performed on the same water area are presented. The results from both of them are presented together on similar types of plots, which allowed for the reliable evaluation of the results and showed the influence of the setup used on the data quality. First, the data from the 1 Hz setup campaign were analyzed.

In the analysis of the results, the internal and external accuracies were verified. The results are shown in Figure 8, including the combined routes in both directions and measurements at land-based stations along the river. The land-based measurements were reduced to the height of the marine gravimeter using free-air reduction [23]. Figure 9 shows the distribution of the differences between the two survey lines. To estimate the internal consistency of the data (internal accuracy), we examined the differences between the results obtained on both survey lines. The internal error was calculated as the mean square from the difference between the two survey lines. 

The estimated internal error of the measurement data was 1.13 mGal. The varying differences between the measurement data on the two lines are plotted in Figure 9. The next step was to calculate the external accuracy by comparison with measurements conducted on the riverbank.

Figure 10 depicts the difference between the average measurements on the two lines and on land. An analysis of thew Odra riverbank measurement differences allowed us to estimate the external consistency of the data. The values of the differences between the points acquired by a land-based gravimeter and those from a marine gravimeter are shown in Figure 11. Because gravimetric measurements had not been conducted before on the Odra, no historical data existed for comparison. Therefore, the external accuracy was verified by comparing measurements made on the riverbank. Based on those results, the calculated external accuracy value was 0.61 mGal. The accuracy was calculated as the root-mean-square (RMS) values of the differences, similarly to the internal accuracy.

Figure 12 presents data from the second campaign, with the usage of a higher (20 Hz) registration frequency. Once again, gravity was measured on the river during two passages, called lines 1 and 2. These lines overlapped each other, so the differences in values between them could be used as the measure of misclosure (internal accuracy). The green crosses in Figure 12 indicate the points where land gravity data were available. The height difference caused by the pier height was reduced using free-air reduction.

Figure 13a depicts the differences both between the lines and between the land data. The RMS misclosure between the lines was 0.0818 mGal. 

The differences between the data collected on the river and the data collected on the land are plotted in Figure 13b. Based on those results, the calculated external accuracy value was 0.2 mGal. The accuracy was calculated as the RMS values of the differences.

After this analysis, the data from all the measurements were used to create the final plot (Figure 14). The data from the second campaign showed significantly fewer fluctuations, which was reflected in the lower internal and external error estimates.

The data from the measurement line provided a basis for calculating the dynamic height on the section of the river covered by the measurements. The value of the dynamic height [23] within the area of measurements was calculated from DGPS and gravimetric results. The difference in dynamic height, ∆*H_dyn_*, between Points A and B can be calculated from Equation (3).
(3)$$∆H_{\mathrm{dyn}}=\int_{A}^{B}dn+\int_{A}^{B}\frac{g-\gamma_{45}}{\gamma_{45}}dn$$

In Equation (3), *d_n_* denotes the height increments between intermediate points, forming a sequence between Points A and B. The second term of the formula is a dynamic correction, depending on the values of the acceleration of gravity *g* at intermediate points and on the value of the reference acceleration *γ_45_*. In this case, the adopted reference acceleration was normal acceleration for the geodetic latitude of 45 degrees.

The reference height was adopted as the height obtained during the binding of the gravimeter still value to the pier point.

Figure 15 presents the distribution of the dynamic height values in the surveyed area.

The data analysis led us to conclude that it is possible to obtain gravimetric data of the gravity distribution on a river with an accuracy comparable to that characterizing marine campaigns. Consequently, such data can be used for calculating the distribution of the dynamic height on rivers using the methodology adopted in this article.

The obtained variation in dynamic height on the river is not large, as predicted for this area. The surveyed river section is located close to the river mouth, so the topography of the surroundings varies only slightly.

The analysis of the data reveals that the changes in the operation frequency of the gravity registration system can lead to significant changes in data quality. For this reason, we have to sum up the results separately.

During the first campaign, it should be noted that the estimated internal accuracy of the gravimetric measurements was greater than the estimated external accuracy. As mentioned before, vessel traffic on the river caused some disturbances. This observation is confirmed by the graphs of the mean amplitude of the acceleration in Figure 6. The differences between the signals from both measurement lines recorded on the survey vessel reached the greatest value of about 2 mGal in places with strong horizontal acceleration. In the area of land-based measurements, no significant disturbances induced by passing other vessels occurred, which decreased the differences between the values of gravity recorded on land and those recorded on the vessel. In addition, it should be noted that the value of the averaged gravity on both profiles was used in the external accuracy analysis. This allowed us to eliminate the residual errors of the Eotvos correction (the movement direction on the two tracks was opposite, and the speeds, nearly identical). Taking into account the above facts, it should be concluded that the error estimation for external accuracy was more reliable than that for internal accuracy.

During the second campaign, the environmental conditions were very similar to those during the first campaign. The traffic on the river was also significant. Despite this, the data collected during the second campaign were characterized by much lower errors, both internal and external. The data processing in the second campaign compared to the first was much more resistant to the noise and disturbances caused by the environmental conditions due to the outlier data. The track on which the measurements were taken was only a few kilometers long. In such conditions, the periods of time in which no external disturbances occurred were relatively short. It was crucial to obtain as many valid data as possible in this short period of time, so the sampling frequency of the whole system and proper outlier data removal were the key features.

## 4. Conclusions

This article examines the dynamic height determined from gravimetric data in an area where vertical accuracy is crucial. As in other European countries, the use of physical heights in river navigation has become more important. The determination of dynamic height requires accurate and reliable data on the distribution of the gravity field in the area concerned. Gravimetric surveys conducted on board ships are an effective method of acquiring detailed local data [24]. The accuracy of the recorded data depends largely on the external conditions prevailing during a survey [25].

The proposed measurement technique allows us to make gravimetric measurements on water bodies where access using land techniques is limited or impossible. This technique can be also used in near-zero-depth coastal zones, which is important for filling the gap between marine and land gravimetric measurements.

The results suggest that river estuary mapping can be performed using a marine gravimetric system with a gimbal sensor. The curvatures of European rivers insignificantly affect survey results. The quality of the data obtained does not differ considerably from that achieved during survey campaigns carried out on much bigger vessels at sea.

The study showed that several possible error sources have to be considered in the determination of dynamic height. Most of these errors seem to have a small impact on the results obtained, and almost all can be minimized by the careful planning of measurements, high-speed registration and careful processing. In the case of our campaign, a higher (20 Hz) frequency of the registration allowed us to achieve better results. This is because by increasing the sampling rate, we could collect more valid data during the period of time when external disturbances were smaller and the recording was less distorted.

We analyzed the possibility of calculating the dynamic height of a river, which is the outcome of this study. Carefully applied, the method of dynamic height determination seems to be a promising technique for examining some current circulation phenomena, at least on large rivers. The method seems to be more useful for determining surface current patterns and speed than for sub-surface water flow systems. The use of this method on the Odra River produced logical and reasonable results, internally consistent and in line with data collected by land-based gravimeters.

Small research vessels can be stable platforms for gravimetric measurements if their centers of gravity (COGs) are sufficiently low. A good example of such a vessel is the Hydrograf XXI, which is a boat with hybrid propulsion (electrical and gasoline engines). The batteries lower its COG and give it additional mass, which is important for dampening hull oscillations. 

The next logical step would be to extend the scope of the dynamic height surveys with the use of this method to cover the entire navigable part of the Odra estuary. This would allow for better predictions of depth distribution on the river at a specific water level, which are essential for the optimal use of the river water resources.

## Figures and Tables

**Figure 1 sensors-20-06044-f001:**
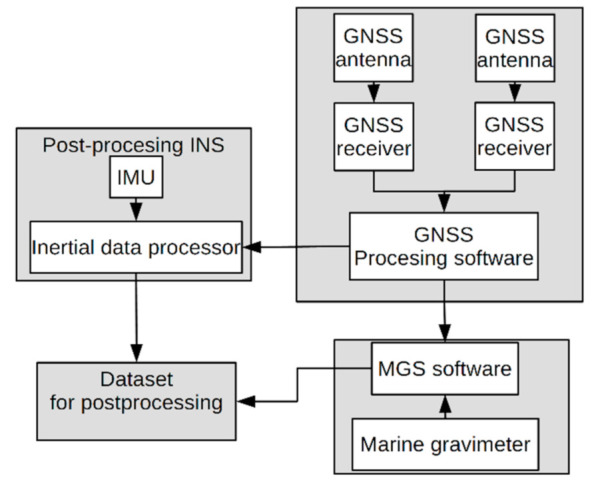
The scheme showing the flow of data during the measurement campaign. Abbreviations used: Global Navigation Satellite Systems (GNSS), Inertial Navigation System (INS), Inertial Measurement Unit (IMU), Marine Gravity System (MGS).

**Figure 2 sensors-20-06044-f002:**
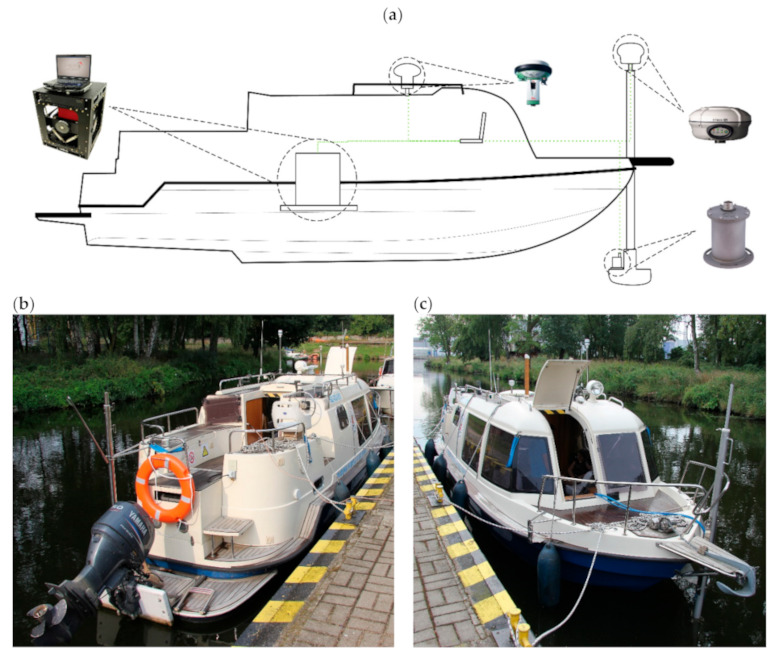
(**a**) Locations of the sensors on the research vessel, from left to right: MGS-6, GS 15, R8 and IMU-10. (**b**) The research vessel during preparation for the campaign: view from stern; (**c**) view from bow.

**Figure 3 sensors-20-06044-f003:**
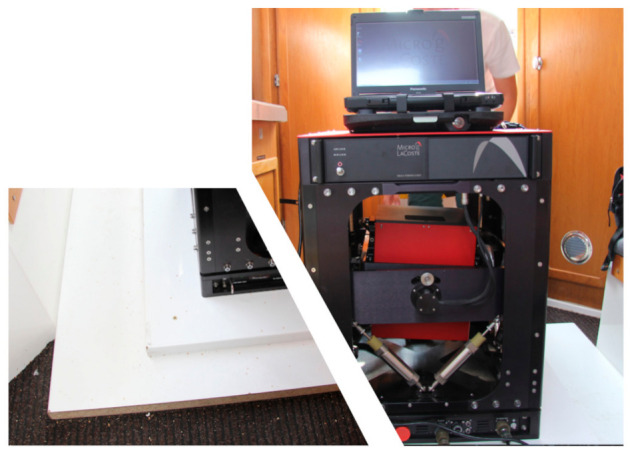
MGS-6 mounted on the survey vessel. On the left is shown how the frame was fitted to the deck using thick chipboard plates.

**Figure 4 sensors-20-06044-f004:**
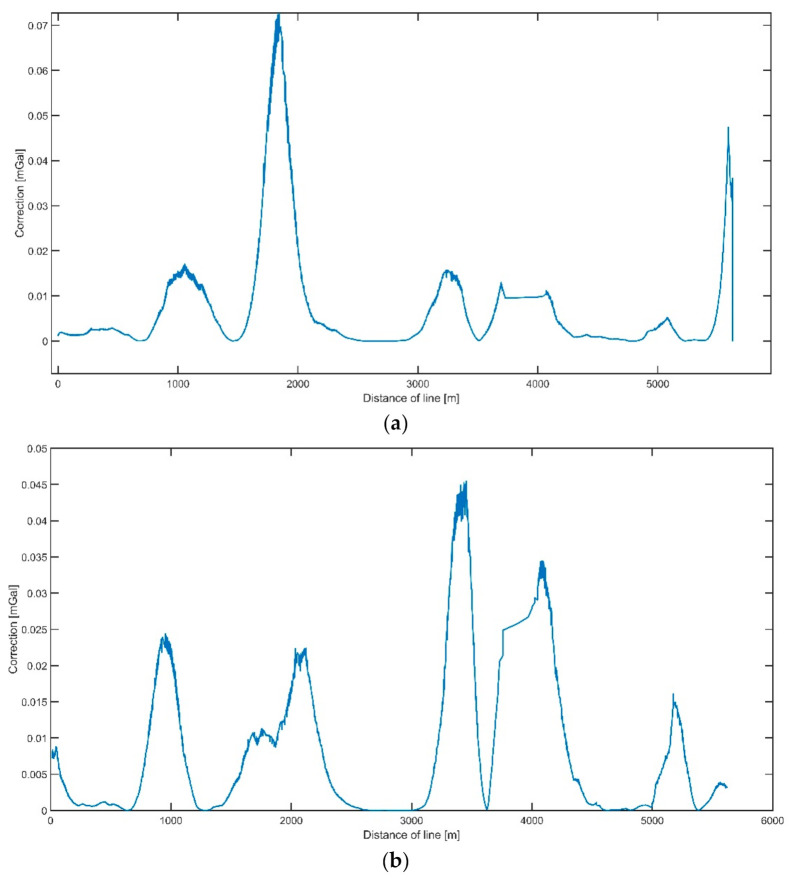
The distribution of the track curvature correction on the measurements of the first line (**a**) and second line (**b**).

**Figure 5 sensors-20-06044-f005:**
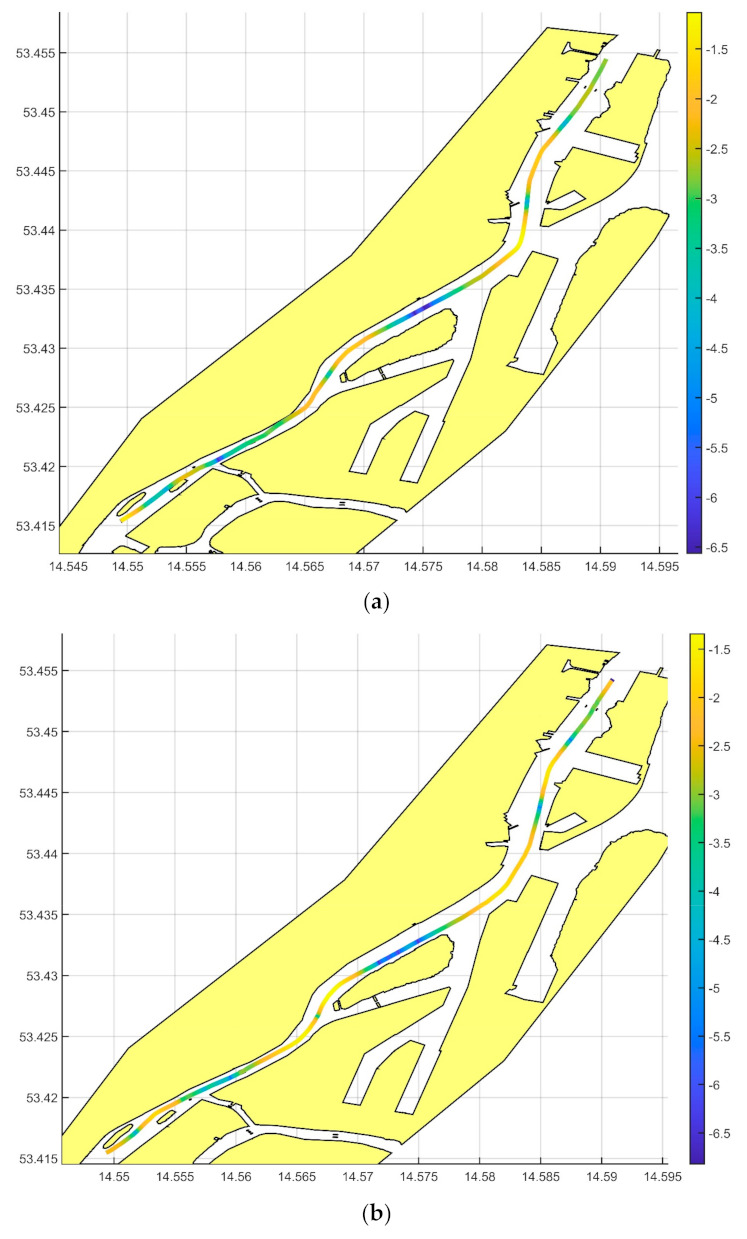
Impact of the curvature of the measurement profile on the recorded gravimetric signal. The spatial distribution of the correction is shown on the maps for the first line (**a**) and for the second line (**b**); a logarithmic scale is used.

**Figure 6 sensors-20-06044-f006:**
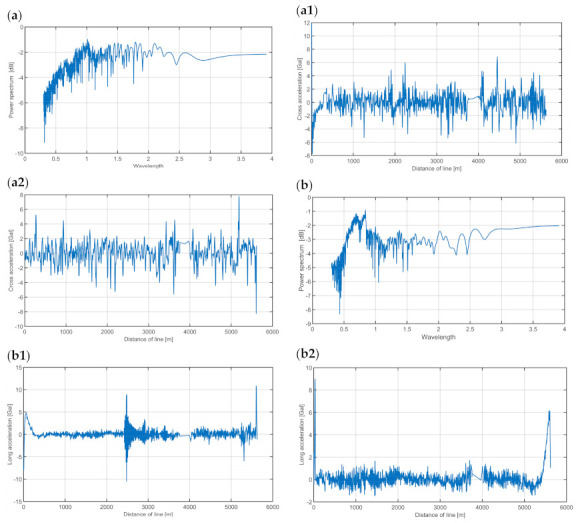
Signals from dilatometers installed in MGS-6. (**a**) Spectral power density for dilatometer signals for cross direction; (**a1**) values of cross accelerations of the first profile; (**a2**) values of cross accelerations of the second profile; (**b**) spectral power density for longitudinal direction; (**b1**) values of longitudinal accelerations for the first profile; (**b2**) values of longitudinal accelerations for the second profile.

**Figure 7 sensors-20-06044-f007:**
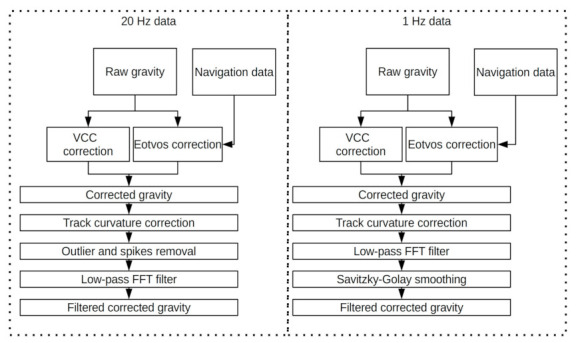
The block scheme showing the data processing routes for 20 Hz and 1 Hz gravimetric data. The FFT refers to Fast Fourier transform.

**Figure 8 sensors-20-06044-f008:**
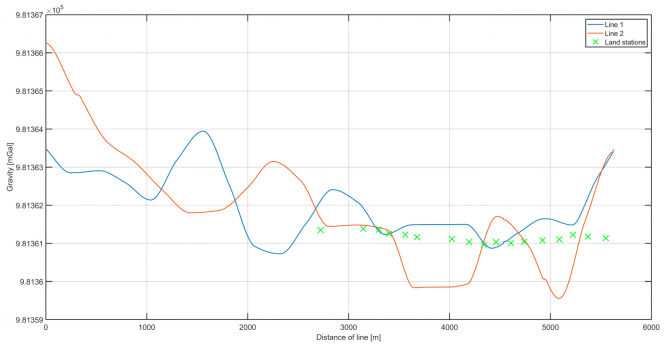
The superimposed values of gravitational acceleration on the two survey lines. In addition, land measurement points are marked, with the obtained values reduced to the height of the dynamic gravimeter.

**Figure 9 sensors-20-06044-f009:**
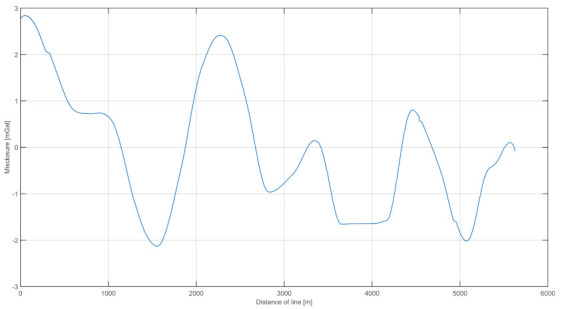
The distribution of differences in the values of gravitational acceleration between the survey lines.

**Figure 10 sensors-20-06044-f010:**
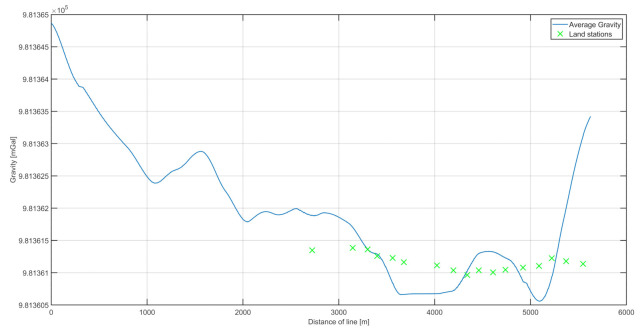
The mean measurements on the river, and data recorded at land-based measuring points.

**Figure 11 sensors-20-06044-f011:**
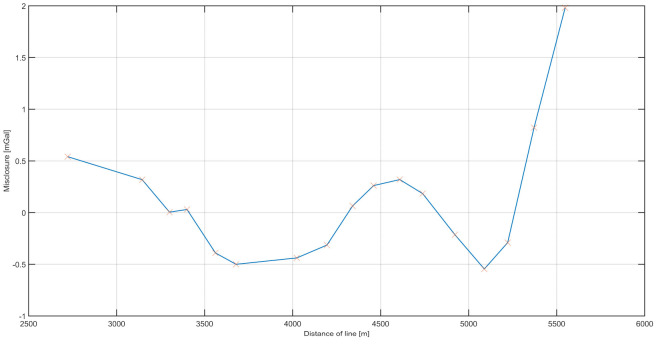
Distribution of differences between the gravity obtained on land and that measured on a survey line. The height differences have been reduced.

**Figure 12 sensors-20-06044-f012:**
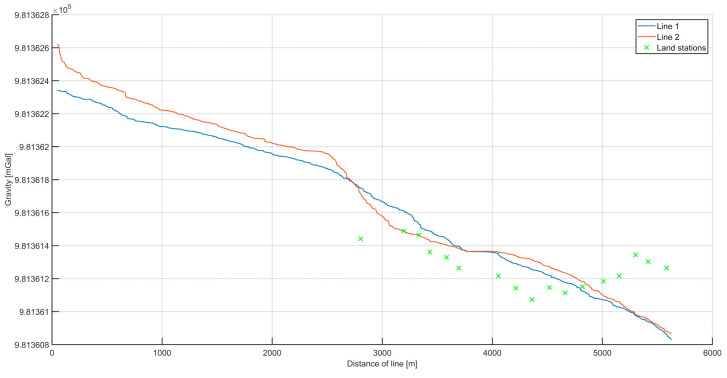
The superimposed values of gravitational acceleration on the two survey lines. In addition, land measurement points are marked, with the obtained values reduced to the height of the dynamic gravimeter. The data presented were processed as a result of the 20 Hz campaign.

**Figure 13 sensors-20-06044-f013:**
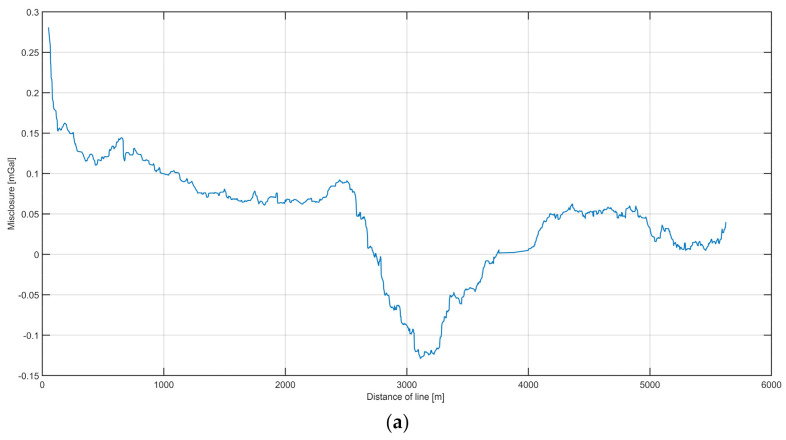
Misclosure analysis of repeated campaign: (**a**) The distribution of differences in the values of gravitational acceleration between the survey lines. (**b**) Distribution of differences between the gravity obtained on land and that measured on a survey line. The height differences have been reduced.

**Figure 14 sensors-20-06044-f014:**
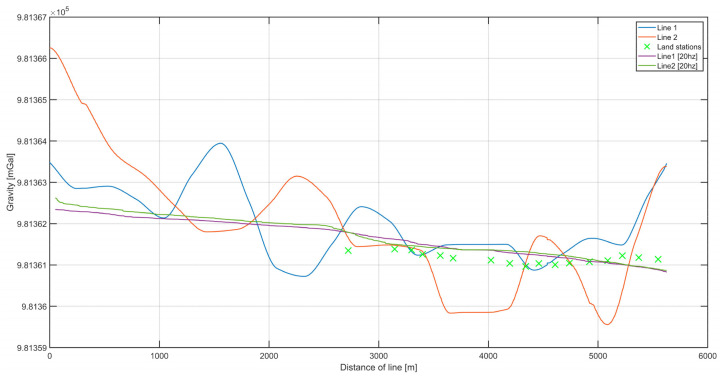
The comparison of results from all campaigns performed on the Odra river.

**Figure 15 sensors-20-06044-f015:**
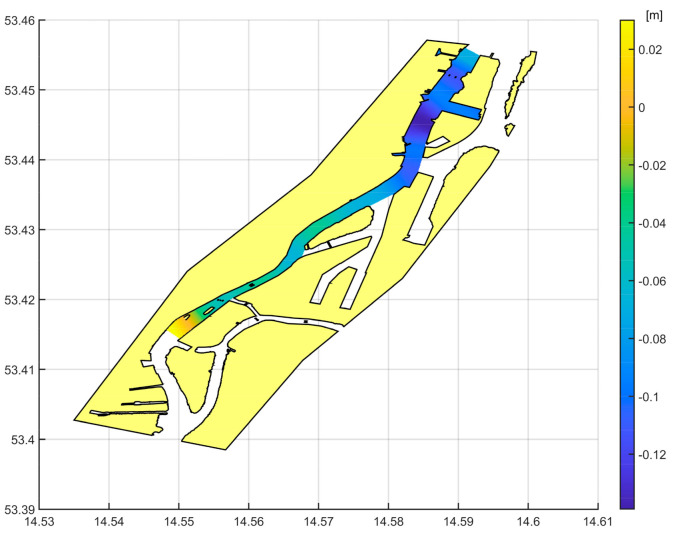
The distribution of dynamic height on the river.

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
