# Peer review of "Analysis of the Dynamic Height Distribution at the Estuary of the Odra River Based on Gravimetric Measurements Acquired with the Use of a Light Survey Boat—A Case Study"

_sensors, 2020, doi:10.3390/s20216044_

Round 1

Reviewer 1 Report

Paper focused on the possible applications of a dynamic gravity meter MGS-6 (Micro-g 13 LaCoste) for determining the dynamic height along the Odra River.

This paper constitutes an interesting application of the gravimetric technique for the purpose to evidence the dynamic height. Results are interesting but improvements are request

  • Abstract needs to be more concise, focus on location, need for investigation methods used expected outcomes.
  • Introduction: authors should be clear how your work could be advances the state of the art or the knowledge in this field. I think that the Authors could be focused this paragraph to specify the peculiar aspects of novelty of this paper with respect to other studies performed from other authors.
  • Authors can provide a paragraph with more information about the methodology.
  • I believe that would be much more informative if Authors provide additional details about the survey strategy and the processing procedure.

Suggest following break down:

  1. a) raw data initial interpretation
  2. b) Modelling description and outcomes
  3. c) processing steps used and why?
  4. d) final interpretation of processed results

Author Response

We acknowledge and thank the reviewer for the valuable comments and suggestions to our manuscript. This has led to major revisions that enhance the quality of our results, in the best interest of the journal’s readership. A detailed list of changes introduced in response to the specific issues raised by the reviewer is presented below. All have the revisions have been underlined in the main manuscript. 

This paper constitutes an interesting application of the gravimetric technique for the purpose to evidence the dynamic height. Results are interesting but improvements are request

Answer:

Great, thank you for the positive feedback.

  • Abstract needs to be more concise, focus on location, need for investigation methods used expected outcomes.

Answer:

The abstract was extensively rewritten to be more informative for the readers. As a result, the nearly whole paragraph is new.

  • Introduction: authors should be clear how your work could be advances the state of the art or the knowledge in this field. I think that the Authors could be focused this paragraph to specify the peculiar aspects of novelty of this paper with respect to other studies performed from other authors.

Answer:

Some parts of the introduction were rewritten; information about other authors similar work was added. Please refer to the indication of the change.

  • Authors can provide a paragraph with more information about the methodology. 
  • I believe that would be much more informative if Authors provide additional details about the survey strategy and the processing procedure.

Answer:

We improved the presentation of the data processing by addition of the flow chart showing the stages of processing (figure 7 corrected version). The material and methods section was partly rewritten. Please refer to the indication of the change. We also improved the description of the experimental setup; additional schemes were added (figures 1 and figure 2a).

Reviewer 2 Report

The work is interesting - the authors used an instrument, which is always applied in the sea survey, on a light survey boat for the river survey. But from my point of view, the paper is not well organized. The cultivation is too flat and reads like a technical note rather than a research paper.

Major revisions.

Some detailed comments:

1. Why you use "the authors", "the research team" in the manuscript? Why not just use "we"?

2. Figure 1 and 2 looks like just "pissed for a shot". Figure 3 left, I don't know what do the authors want to show using this photo. At least you should provide a reference for scale of the boat or just mark the length on it. I recommend you select other photos and combine them into one figure.

3. Figure 6, show them in 2*3 (two rows, three columns) or 3*2 forms.

4. In the Introduction, I want to clearly see "why" you are doing this work, and what is the objective? Also, state-of-the-art is missing in this part.

5. Section Two is poorly organized. You may need a flow chart to show the data processing method, and several sub-sections could be better in this part.

6. Results part, please rewrite this part not just show the figures: what are your points? - as I cannot find your main contributions and innovations.

7. I cannot get your conclusions.

The experiment is good, and the application is potential for publication. But they are not organised as a research paper. I encourage the authors to learn how to write scientific papers.

Author Response

We acknowledge and thank the reviewer for the valuable comments and suggestions to our manuscript. This has led to major revisions that enhance the quality of our results, in the best interest of the journal’s readership. A detailed list of changes introduced in response to the specific issues raised by the reviewer is presented below. All have the revisions have been underlined in the main manuscript. 

The work is interesting - the authors used an instrument, which is always applied in the sea survey, on a light survey boat for the river survey.

Answer:

Great, thank you for the positive feedback.

But from my point of view, the paper is not well organized. The cultivation is too flat and reads like a technical note rather than a research paper.

Major revisions.

Some detailed comments:

  1. Why you use “the authors”, “the research team” in the manuscript? Why not just use “we”?

Answer:

We changed “the research team” to the “we” where it was possible. Although, “the authors” is the common form in scientific writing.

  1. Figure 1 and 2 looks like just “pissed for a shot”. Figure 3 left, I don’t know what do the authors want to show using this photo. At least you should provide a reference for scale of the boat or just mark the length on it. I recommend you select other photos and combine them into one figure.

Answer:

The requested figures were corrected. The new version is arrangement is better; the new comments and caption to the figures make sense of presentation clearer.

  1. Figure 6, show them in 2*3 (two rows, three columns) or 3*2 forms.

Answer:

Figure 6 was rearranged to the 3*2 form. The readability of each subplot was enhanced (in order to keep it readable despite smaller size).

  1. In the Introduction, I want to clearly see “why” you are doing this work, and what is the objective? Also, state-of-the-art is missing in this part.

Answer:

The Introduction was partly changed. Especially its ending was rewritten to indicate the main goal of the experiment.

  1. Section Two is poorly organized. You may need a flow chart to show the data processing method, and several sub-sections could be better in this part.

Answer:

The organization of section two was improved. We add more details about the equipment used and how it was placed on the vessel—two flow charts where added: one describing the data collection process, another for data processing routes.

  1. Results part, please rewrite this part not just show the figures: what are your points? - as I cannot find your main contributions and innovations.
  2. I cannot get your conclusions.

Answer:

In our opinion, this is caused by some mess et the end of the first version of the article. Thank you for pointing this out. We rewrote the last section in order to improve the visibility of the conclusion, so the reader should not be confused anymore.

Round 2

Reviewer 1 Report

Paper was improved and merit to be published